# The Association of Mobile Health Applications with Self-Management Behaviors among Adults with Chronic Conditions in the United States

**DOI:** 10.3390/ijerph181910351

**Published:** 2021-09-30

**Authors:** Hao Wang, Amy F. Ho, R. Constance Wiener, Usha Sambamoorthi

**Affiliations:** 1Department of Emergency Medicine, JPS Health Network, Fort Worth, TX 76104, USA; hwang@ies.healthcare (H.W.); amyho@ies.healthcare (A.F.H.); 2Department of Dental Practice and Rural Health, West Virginia University, Morgantown, WV 26506, USA; rwiener2@hsc.wvu.edu; 3Texas Center for Health Disparities, Department of Pharmacotherapy, College of Pharmacy, University of North Texas Health Science Center, Fort Worth, TX 76107, USA

**Keywords:** mHealth apps, self-management behaviors, chronic diseases

## Abstract

Background: Mobile applications related to health and wellness (mHealth apps) are widely used to self-manage chronic conditions. However, research on whether mHealth apps facilitate self-management behaviors of individuals with chronic conditions is sparse. We aimed to evaluate the association of mHealth apps with different types of self-management behaviors among patients with chronic diseases in the United States. Methods: This is a cross-sectional observational study. We used data from adult participants (unweighted *n* = 2340) of the Health Information National Trends Survey in 2018 and 2019. We identified three self-management behaviors: (1) resource utilization using electronic personal health records; (2) treatment discussions with healthcare providers; and (3) making healthcare decisions. We analyzed the association of mHealth apps to self-management behaviors with multivariable logistic and ordinal regressions. Results: Overall, 59.8% of adults (unweighted number = 1327) used mHealth apps. Adults using mHealth apps were more likely to use personal health records (AOR = 3.11, 95% CI 2.26–4.28), contact healthcare providers using technology (AOR = 2.70, 95% CI 1.93–3.78), and make decisions on chronic disease management (AOR = 2.59, 95% CI 1.93–3.49). The mHealth apps were associated with higher levels of self-management involvement (AOR = 3.53, 95% CI 2.63–4.72). Conclusion: Among individuals with chronic conditions, having mHealth apps was associated with positive self-management behaviors.

## 1. Introduction

Self-management plays a central role in improving clinical outcomes among patients with chronic conditions [1,2]. In general, self-management includes six behaviors: problem-solving, decision making, resource utilization, patient and provider communication, action planning, and self-tailoring [3]. These self-management behaviors, either alone or in combination, have improved patient clinical outcomes [3,4]. Therefore, interventions have been developed and administered to enhance self-management behaviors. Previous interventions have typically focused on in-person educational sessions or the distribution of printed materials [1,3,5,6]. For example, the Chronic Disease Self-Management Programs (CDSP) promoted physical activity, stress reduction, and a healthy diet through goal setting and action-taking [7,8]. In recent years, with the rise of the internet and health information technology (HIT), other interventions, including mobile health (mHealth) for the self-management of chronic conditions, have been investigated [9,10].

The World Health Organization’s Global Observatory for eHealth (GOe) defined mHealth in 2011 as “medical and public health practice supported by mobile devices, such as mobile phones, patient monitoring devices, personal digital assistants (PDAs), and other wireless devices” [11]. With the emergence of mHealth practice, certain self-management behaviors have been reported to improve chronic disease care through mHealth apps [12,13]. These self-management behaviors mainly include self-monitoring and action planning. For example, a mHealth app for gout management improved patient daily pain monitoring and long-term gout self-care behavior [14]. Patients with diabetes use mHealth apps for serum glucose tracking and monitoring, which have been reported to improve diabetes clinical outcomes [15]. In another diet control study, 60% of patients used mHealth apps for their diet control/monitoring [16]. In general, mHealth apps that facilitate tracking the progress of goals set (action planning and self-tailoring) by the individual have been well-studied [14,17,18,19]. However, thorough analyses of using mHealth apps affecting the other four reported self-management behaviors (e.g., problem-solving, decision-making, resource utilization, and patient–provider communication) are less studied and remain largely unknown.

An ideal and effective intervention should promote a variety of self-management behaviors. The use of the mHealth app is undoubtedly a useful tool, especially in the current era. The coronavirus disease pandemic has made in-person interventions for promoting self-management behaviors very challenging [20,21]. Recognizing the role of the mHealth app in relation to the variety of self-management behaviors could serve as a foundation for future implementation of the non-in-person intervention (e.g., advocating mHealth use). Currently, Health Information National Trend Surveys (HINTS) collect nationally representative data from American adults aged 18 or older to assess the impact of the HIT, including the report of mHealth app use and the status of individual self-management behaviors [22]. To better understand the role of using the mHealth app affecting an individual’s different self-management behaviors, it is ideal for analyzing the use of the mHealth app in association with each self-management behavior separately. However, one of the self-management behaviors, problem-solving, by definition, may involve multiple steps with overlapping other self-management behaviors (such as recognizing problems, finding ways to solve problems, and finally taking actions), thereby hard to investigate its direct association. Therefore, in this study, we examined the association of mHealth app use to three different patient self-management behaviors (resource utilization through the use of electronic personal health records; patient–provider communication; and decision-making) among adults with chronic conditions by analyzing data from this nationally representative survey.

## 2. Materials and Methods

### 2.1. Study Design and Data Source

Health Information National Trend Survey (HINTS) is a nationally representative cross-sectional survey administered by the National Cancer Institute with publicly available data [22]. We conducted a secondary data analysis from HINTS 5 Cycle 2 and 3 that were performed before the coronavirus pandemic (i.e., 2018 to 2019). The target population of the HINTS is all adults in the civilian non-institutionalized population of the United States [22]. The survey was randomly assigned to a US address included in the Marketing Systems Group (MSG) database. An eligible participant was selected using the next birthday method (if a household has more than one eligible participant, the responder is assigned to the person in the household whose birthday is next). While HINTS 5, cycle 2 (2018) consisted of a single-mode mail survey. HINTS 5, Cycle 3 (2019) was administered using a mail survey and a push-to web pilot (Web Pilot)) [22]. Due to the nature of de-identified data, this study was exempted from review by the regional Institutional Review Board (IRB, No.1705528-1). We used the STROBE (Strengthening the Reporting of Observational Studies in Epidemiology) as our reporting guideline [23].

### 2.2. Study Participants

We restricted our analysis to adults with any of the following chronic conditions: (1) diabetes, (2) hypertension, (3) heart diseases, and (4) lung diseases. These conditions were chosen because of their high prevalence, the need for and benefits from everyday self-management, and common challenges associated with self-management. In addition, we excluded individuals with no internet connections or no smartphone(s) or tablet(s). We further excluded participants who had missing data on key variables (self-management behaviors, mHealth app, age, sex, and race/ethnicity). Details of the number of adults excluded are shown in Figure 1.

### 2.3. Measures

#### 2.3.1. Self-Management Behaviors

Across all chronic conditions, six self-management behaviors [3] have been proven to improve health outcomes [1,2]. These six behaviors are: (1) patient–provider communication; (2) resource utilization; (3) decision-making; (4) problem-solving; (5) action planning; and (6) self-tailoring. As the adoption of mHealth apps related to action planning and self-tailoring (example: tracking and reaching the goal) have been well studied [12,14,19,24], we did not report in this study but included analysis results in the Appendix A. HINTS 5 Cycle 2 and 3 survey did not capture problem-solving as a separate self-management behavior, so it is hard to analyze independently [22]. Therefore, in this paper, we included only three self-management behaviors.

Patient–provider communication. This was assessed with the yes/no response to the question: “Has your tablet or smartphone helped you in discussions with your health care provider?” [22].Resource utilization. This was assessed with a yes/no dichotomy of the personal health record use question: “How many times did you access your online medical record in the last 12 months?” [22]. If the respondents answered “none”, then they were considered as non-using resources. Individuals who accessed their online medical records at least once were considered to have utilized resources.Decision-making. This was assessed with the yes/no response to the question: “Has your tablet or smartphone helped you make a decision about how to treat an illness or condition?” [22].

We further divided individuals into four groups to indicate levels of self-management behaviors. Since all these three self-management behaviors are considered patients’ healthcare engagement, we further categorized different levels of healthcare engagement based upon their different self-management behavior involvement [25,26,27,28]. Individuals who answered “no” to all three self-management behaviors were categorized as a “no engagement” group. Individuals who answered “yes” to one self-management behavior were categorized as “low level of engagement”. Individuals with two self-management behaviors were categorized as “moderate level of engagement”. Ones with all three self-management behaviors were categorized as “high level of engagement”.

#### 2.3.2. Key Independent Variable: Use of mHealth Apps

We created an indicator variable (yes/no) to capture the use of mHealth apps based on the following HINTS question: “on your tablet or smartphone, do you have any apps related to health and wellness?” [22]. Individuals who answered “yes” were considered as having mHealth apps, and ones who answered “no” were considered as not having mHealth apps in their tablet(s) or smartphone(s). Since this is a key independent variable, we excluded ones with missing/in error/inapplicable answers and excluded ones with “don’t know” answers.

#### 2.3.3. Other Explanatory Variables

We included biological and socioeconomic variables: age, sex, race/ethnicity, insurance level, household income levels, and individual education levels. We used: standard five-level age categories (i.e., 18–34, 35–49, 50–64, 65–74, and 75+); [22] two groups in sex (male and female); four groups in race/ethnicity (Non-Hispanic White, Non-Hispanic Black, Hispanic, and others); two groups in insurance level (yes or no); five groups in household income levels (less than $20k, $20k–<35k, $35k–<50k, $50k–<75k, and $75k+); and four groups in education levels (less than high school, high school or its equivalent, some college, college and above). In addition, individuals with chronic conditions were divided into two groups (one chronic condition versus more than one chronic condition).

#### 2.3.4. Statistical Analysis

Weighted percentages were derived using replicate weights. Significant group differences were tested with Rao–Scott chi-square tests. It is possible that self-management behaviors can be achieved with the use of the mHealth app or other components of HIT such as email and the internet. In this study, we determine the association of mHealth app use with each of the self-management behaviors. The mHealth app may facilitate the use of other HIT components and further encourage self-management behavior. We performed separate multivariable logistic regression analyses for self-engagement behavior. Age, sex, race/ethnicity, insurance, income, and education were adjusted in these analyses. Hosmer-Lemeshow Goodness-of-fit test (GOF) was performed to determine the evidence of model fit with *p* > 0.05 indicating a good model fit. With secondary data, small cell sizes can produce unreliable parameter estimates. To ensure the reliability of estimates, we used relative standard errors (RSE). Smaller RSEs indicate reliable results. We observed that RSEs were less than 25%, suggesting good reliability of our findings.

We also performed an ordinal regression analysis to determine the association between mHealth apps and different levels of self-management behavior. We analyze the probability of belonging to each level by using the “margins” command in STATA [29]. All statistical analysis was conducted by STATA version 14.0 (College Station, TX, USA). We followed the HINTs guidelines for analysis [30] and used survey procedures with replicate weights for all analyses.

## 3. Results

The final analysis included 2340 surveys (weighted population of 45,336,040 in 2018 and 69,292,723 in 2019). Table 1 shows the biological and socioeconomic characteristics of the study population in terms of whether having the mHealth apps in their tablet(s) or smartphone(s). We did not observe differences in sex, race/ethnicity, insurance status, or the number of chronic conditions sustained by individuals regardless of having mHealth apps. However, individuals with mHealth apps tended to be younger, more educated, and had higher income levels (Table 1).

A higher percentage of those with mHealth apps on their smartphone(s) or tablet(s) reported support in making treatment decisions (53.9% vs. 30.9%), discussions with their healthcare providers (53.6% vs. 28.7%), and using electronic personal health records (69.4% vs. 40.6%) compared to those without mHealth apps (*p* < 0.0001, Table 2).

After adjustments for biological and cultural variables (i.e., age, sex, and race/ethnicity), socioeconomic status (income and education), and access to care (health insurance) variables, adults with mHealth apps in their smartphone(s) or tablet(s) were more likely to use personal health records compared to those without mHealth apps. The adjusted odds ratio (AOR) for using personal health records was 3.11 with a 95% confidence interval (CI) of 2.26–4.28 (*p* < 0.001). The Hosmer–Lemeshow goodness-of-fit test showed good model fit (GOF, *p* = 0.9812). The AORs for supporting discussions with healthcare providers and helping to make treatment decisions were 2.70 (95% CI 1.93–3.78, *p* < 0.001, GOF, *p* = 0.3495), and 2.59 (95% CI 1.93–3.49, *p* < 0.001, GOF, *p* = 0.8115) respectively (see Figure 2).

The ordinal regression analyzing the association of mHealth apps on the levels of self-management behavior (i.e., none, low, moderate, and high levels of engagement) revealed similar findings. The proportional OR was 3.53 with a 95% CI of 2.63–4.72 (Table 3, *p* < 0.001). It can be interpreted as, for individuals who had mHealth apps, the odds of a high level of self-management behavior versus moderate + low + no engagement were 3.53 times higher than those without mHealth apps. Likewise, the odds of the combined categories of high and moderate levels of self-management engagement versus low + no engagement were 3.53 times higher among ones with mHealth apps than those without mHealth apps. Similarly, the odds of the combined categories of high + moderate + low versus no engagement were 3.53 times higher among ones with mHealth apps than those without (Table 3).

We also computed the adjusted probability of belonging to levels of self-management behaviors. The probability of “none” was higher among individuals without mHealth apps (0.37, 95% CI 0.32–0.42) than those with mHealth apps (0.15, 95% CI 0.12–0.18, *p* < 0.0001). On the other hand, the probability of having high engagement was higher among individuals with mHealth apps (0.31, 95% CI 0.27–0.34) than those without (0.11, 95% CI 0.09–0.14, *p* < 0.0001). Overall, based on our model (Table 3), an individual who had mHealth apps was more likely to be classified as having moderate to high levels of self-management behavior and least likely to be classified as no or low levels of self-management behavior.

## 4. Discussion

Self-management of chronic conditions by individuals has become an integral part of comprehensive chronic disease management, improves health outcomes, and reduces healthcare expenditures [1,2]. Although self-management education and interventions are implemented with in-person visits, self-management with mHealth apps has been widely practiced. In this study, we investigated the role of mHealth apps in facilitating three self-management behaviors. The use of mHealth apps is not new. The mHealth apps have been used to record individuals’ vital signs (e.g., blood pressure) and monitor day-to-day health activities/goals (e.g., daily exercise, diet calories) [19,31,32]. However, the potential benefits of mHealth apps in promoting other self-management behaviors have rarely been reported. This study extended the link from mHealth apps to different self-management behaviors (i.e., resource utilization through personal health records use, communication with providers, and help decide on health management).

We observed that mHealth apps on smartphones or tablets facilitated these self-management behaviors. With the rapid development of the internet and mobile platforms, the development and use of mHealth apps to manage health and wellness have increased [9,33]. [34,35] Previous studies showed younger age, persons with high education and income levels were more likely to use mHealth apps for the self-management behaviors studied [24,36]. Our findings are consistent with previous reports. In addition, we focused on a particular population (i.e., individuals with chronic conditions) and found that approximately 59.8% of individuals with chronic conditions had mHealth apps on their tablets or smartphones. The number of individuals with chronic conditions who use mHealth apps has increased from 45.8% in 2017 [24] (HINTS 5 Cycle 1) to 59.8% in 2019 (HITNS 5 Cycle 2 and 3) [22].

We observed that mHealth apps were positively associated with resource utilization (i.e., personal health records). The use of personal health records via patient portals can empower patients to track and assess health, track progress towards health goals, manage health between visits, schedule clinic appointments electronically, and increase e-communications between patients and providers [37,38,39]. Personal health record use has been shown to improve patient clinical outcomes in some studies [40,41]. Thus, any steps taken to increase the use of personal health records (in our case, mHealth app use) can lead to improved outcomes [40,41].

Patient–provider effective communication, another key domain of self-management behavior, plays a central role in chronic disease management [2,42] and health outcomes [43]. It is quite challenging for healthcare providers alone to manage patients with chronic diseases. Communication with the provider is critical during the current coronavirus pandemic which has limited in-person contact with providers. While telemedicine and electronic communication (e-communication) with healthcare providers have become extremely valuable. [20,44], strategies to enhance telemedicine and e-communication are needed. In this context, mHealth apps supporting e-communication with healthcare providers can open another valuable intervention for healthcare promotions.

Overall, shared decision-making between healthcare providers and patients is recommended [45,46]. Studies also confirmed the benefit of shared decision-making on chronic disease management. However, it is always challenging to empower and educate individuals to make decisions. mHealth apps may prove to be a valuable tool in increasing patients’ knowledge about their disease(s) and self-management of the disease(s). For example, in a study of individuals with diabetes, mHealth apps were beneficial in tracking glucose control and providing diabetes education [47,48]. Using mHealth apps is a relatively easy way to expand an individual’s disease-related health knowledge, thus helping with appropriate decision making—one of the six domains of self-management behavior.

Previous studies reported that focusing on just one or two self-management behaviors might not be optimal for ideal chronic disease control [31,49]. Such findings raised the question of different engagement levels of self-management behavior. Our findings suggest that having mHealth apps is associated with a moderate-to-high engagement level of self-management, thereby including diverse domains of self-management behaviors. Optimal healthcare outcomes might thus be reached with higher levels of self-management involvement. However, we understand our findings only show the association between mHealth apps and certain self-management behaviors. A direct link between mHealth apps and healthcare outcomes cannot be established. Therefore, future studies to investigate the direct association of mHealth apps related to healthcare outcomes are warranted.

Our study findings have to be interpreted in the context of the study’s strengths and limitations. As we relied on previously collected data, we were limited in the choice and definition of variables. For example, chronic conditions were limited to four available conditions provided in the survey. As the HINTS survey is a cross-section survey, we can only determine the association and cannot assess whether mHealth apps use caused better adoption of self-management behaviors (i.e., causality). In addition, multiple factors could affect individuals’ self-management behaviors. In this study, we only adjusted a limited set of factors. Other factors (e.g., internet access, enrollment in lifestyle intervention programs, etc.) might potentially affect the use of the mHealth app and self-management behaviors. Therefore, future studies are warranted to investigate other potential risks affecting individuals’ self-management behaviors. From this study, we cannot determine if having mHealth apps on smartphones and tablets translates into “meaningful” use of the apps. However, other HINTS data suggest that an overwhelming majority of individuals with mHealth apps use them. Therefore, a large-scale prospective study is needed to determine the benefit of using mHealth promoting better health outcomes.

## 5. Conclusions

Our study findings support mHealth apps to promote higher-level self-management behaviors among individuals with chronic conditions. However, a prospective interventional study to determine the use of mHealth apps for the optimal chronic disease care outcomes is needed.

## Figures and Tables

**Figure 1 ijerph-18-10351-f001:**
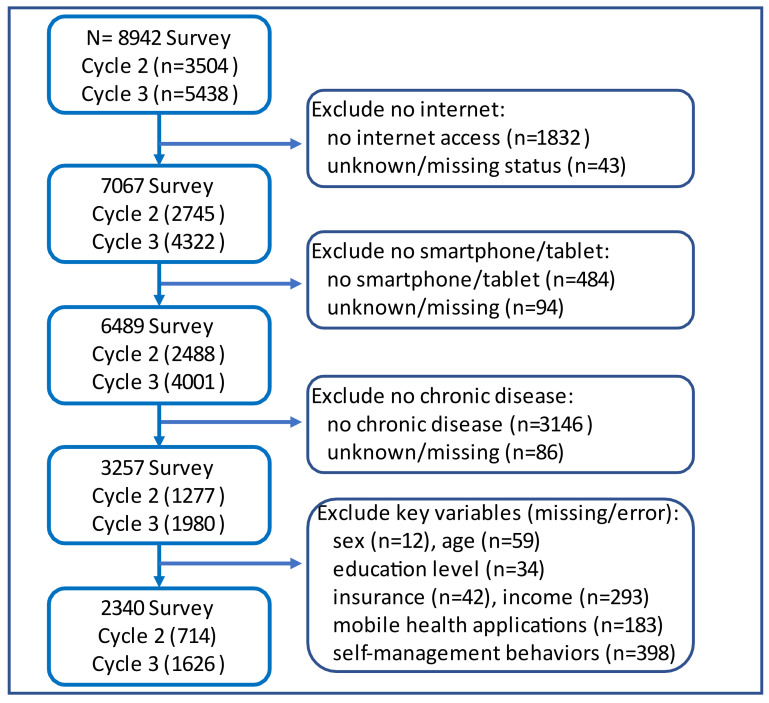
Study Flow Diagram.

**Figure 2 ijerph-18-10351-f002:**
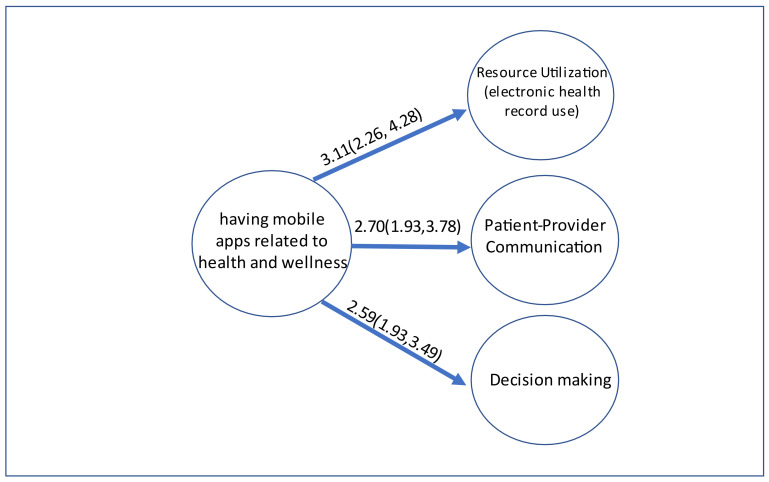
The Adjusted Odds Ratios and 95% Confidence Interval of mHealth Apps Associated with Three Different Management Behaviors Separately by using Multivariable Logistic Regressions.

**Table 1 ijerph-18-10351-t001:** Biological and Socioeconomic Characteristics of the Study Sample by Individuals Having Mobile Health Applications (Health Information National Trends Survey-5, Cycle 2 and 3).

Biological and Socioeconomic Variables		Individual Having mHealth Apps		Individual Not Having mHealth Apps		
		*n*	Wt %	*n*	Wt %	*p*-Value
ALL		1327	59.8	1013	40.3	
Sex						0.0695
	Female	760	52.0	529	46.0	
	Male	567	48.0	484	54.0	
Age in Years						0.0027
	18–34 years	130	19.1	51	11.7	
	35–49 years	255	25.5	128	23.1	
	50–64 years	525	37.8	354	36.2	
	65–74 years	298	12.1	333	20.6	
	75 and older	119	5.5	147	8.4	
Race/Ethnicity						0.5404
	Non-Hispanic White	814	66.4	676	68.4	
	Non-Hispanic Black	191	8.7	115	8.5	
	Hispanic	153	13.3	111	14.4	
	Others	169	11.7	111	8.7	
Education						0.0047
	Less than High School	31	4.0	35	4.6	
	High School or equivalent	229	25.0	271	37.3	
	Some College	331	35.2	266	31.7	
	College and above	736	35.8	441	26.4	
Insurance						0.5819
	Yes	1286	94.6	970	93.5	
	No	41	5.4	43	6.5	
Annual Income						0.0003
	Less than 20k	140	10.3	170	16.5	
	20k–<35k	136	9.1	136	11.8	
	35k–<50k	158	13.1	146	14.8	
	50k–<75k	255	18.1	234	24.3	
	75k+	638	49.4	327	32.6	
Chronic diseases						0.3233
	1 chronic condition	812	65.5	596	61.5	
	>1 chronic conditions	515	34.5	417	38.5	

Note: Based on pooled data of 2340 adult participants without missing data on sex, age, race/ethnicity, education, insurance, and income, from Health Information National Trends Survey 5 Cycles 2 and 3. Significant group differences were tested with Rao–Scott chi-square tests. Weighted Percentages were derived using replicate weights. Wt: weighted.

**Table 2 ijerph-18-10351-t002:** Comparisons of Different Self-management Behaviors Between Individuals with and without mHealth Applications.

Self-Management Behaviors	Individuals with mHealth Apps		Individuals without mHealth Apps		
	*n*	Wt%	*n*	Wt%	*p* Value
Decision Making Yes No	- 725 602	- 53.9 46.1	- 314 699	- 30.9 69.1	<0.0001
Patient–provider Communication Yes No	- 741 586	- 53.6 46.4	- 292 721	- 28.7 71.4	<0.0001
Resource Utilization (electronic health record use) Yes No	- 877 450	- 69.4 30.6	- 441 572	- 40.6 59.4	<0.0001

Note: Based on pooled data of 2340 adult participants without missing data from HINTS 5 Cycle 2 and 3. Significant group differences were derived from Rao–Scott chi-square tests.

**Table 3 ijerph-18-10351-t003:** Having mHealth apps was associated with different engagement levels of self-management behavior.

	Proportional Odds Ratios	95% Confidence Interval	*p* Value
Having mHealth apps No Yes	- reference 3.53	- reference [2.63–4.72]	- - <0.0001
Cutoff	Point	95% Confidence Interval	*p* value
Cut1 Cut2 Cut3	0.28 1.72 2.92	[−0.44, 0.99] [0.99, 2.44] [2.21, 3.63]	0.441 <0.0001 <0.0001
Margins of Different Engagement Levels of Self-management	- probability	- 95% Confidence Interval	- *p* value
No engagement Without mHealth apps With mHealth apps	- 0.37 0.15	- [0.32, 0.42] [0.12, 0.18]	- <0.0001 <0.0001
Low level of engagement Without mHealth apps With mHealth apps	- 0.34 0.27	- [0.30, 0.37] [0.23, 0.30]	- <0.0001 <0.0001
Moderate level of engagement Without mHealth apps With mHealth apps	- 0.18 0.28	- [0.15, 0.21] [0.24, 0.32]	- <0.0001 <0.0001
High level of engagement Without mHealth apps With mHealth apps	- 0.11 0.31	- [0.09, 0.14] [0.27, 0.34]	- <0.0001 <0.0001

Note: Based on pooled data of 2340 adult participants without missing data on sex, age, race/ethnicity, education, insurance, income, mHealth app, and self-management behaviors, from Health Information National Trends Survey 5 Cycles 2 and 3 and separate multivariable logistic regressions of self-management behaviors. The final model adjusted for age, sex, race/ethnicity, income, education, and health insurance.

## Data Availability

Health Information National Trend Survey 5 Cycle 2 and 3 Data is publicly available via http://hints.cancer.gov (accessed on 27 September, 2021).

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
