# Peer review of "The Association of Mobile Health Applications with Self-Management Behaviors among Adults with Chronic Conditions in the United States"

_ijerph, 2021, doi:10.3390/ijerph181910351_

Round 1

Reviewer 1 Report

This is a sound, well written study on a very interesting and relatively novel topic. The AutHors fully ackonwledge the main study limitation i.ee. its cross sectional nature. However the results are very interesting and well discussed

I have two minor suggestions to make:

LN 69-70

Health Information National Trend Surveys (HINTS) is a nationally representative 69 survey administered by the National Cancer Institute with publicly available data.21

PLEASE PROVIDE MORE DETAIL ON HOW THE HINT WAS CONDUCTED AND MORE SPECIFICALLY ON HOW THE TARGET POPULATION WAS SELECTED AND RECRUITED

LN 91-92

As the adoption of mHealth apps related to self-tailoring (example: track- ing) have been well studied,14; 23, we did not analyze self-tailoring behavior.

SINCE YOU HAVE THE INFORMATION I BELIEVE IT WOULD BE INTERESTING TO VERIFY ALSO THIS ASSOCIATION, TO VERIFY WHETHER FINDINGS IN THIS POPULATION ARE CONSISTENT WITH PREVISOUS REPORTS

Author Response

This is a sound, well written study on a very interesting and relatively novel topic. The AutHors fully acknowledge the main study limitation i.ee. its cross-sectional nature. However, the results are very interesting and well discussed

Thank you very much for the positive feedback.

I have two minor suggestions to make:

LN 69-70 Health Information National Trend Surveys (HINTS) is a nationally representative 69 survey administered by the National Cancer Institute with publicly available data.21 PLEASE PROVIDE MORE DETAIL ON HOW THE HINT WAS CONDUCTED AND MORE SPECIFICALLY ON HOW THE TARGET POPULATION WAS SELECTED AND RECRUITED

Response: Thanks for reviewer’s valued comment. We revised and added the followings in the method section (Line 105-114).

The target population of the HINTS is all adults in the civilian non-institutionalized population of the United States.22 The survey was randomly assigned to a US address included in the Marketing Systems Group (MSG) database.  Eligible participant was selected using next birthday method (if a household has more than one eligible participant, the responder is assigned to the person in the household whose birthday is next).  While HINTS 5, cycle 2 (2018) consisted of a single-mode mail survey. HINTS 5, Cycle 3 (2019) was administered using a mail survey and a push-to web pilot (Web Pilot)).”

LN 91-92 As the adoption of mHealth apps related to self-tailoring (example: tracking) have been well studied,14; 23, we did not analyze self-tailoring behavior.

SINCE YOU HAVE THE INFORMATION, I BELIEVE IT WOULD BE INTERESTING TO VERIFY ALSO THIS ASSOCIATION, TO VERIFY WHETHER FINDINGS IN THIS POPULATION ARE CONSISTENT WITH PREVISOUS REPORTS

Response: To address the reviewer’s comment, we conducted additional analyses.   The results from these additional analyses are summarized in the Appendix.  We observed that 69.3% of respondents used mHealth App for self-tracking.  These results are consistent with published reports (Please see the supplementary).

Reviewer 2 Report

Abstract

  • Please provide n=, % in abstract

Introduction

  • Justification of the choice of the three areas of this study (of the six) need to be explained and justified earlier in the paper

Material and Methods

  • You did not conduct a study but selected a cross-sectional slice of the data
  • Flow diagramme is useful
  • Please clarified how some of the variables were operationalised. If you look at the questions - all of them could be done via the Internet email or phone - not necessarily mHealth apps? This present a major confounder in your study as the study associations imply causation but may be due to confounding.
  • Provide motivation for engagement classification
  • Justify age group categorisation
  • In tables please add the Test used and remove the leading 0s on the p-values.  If a value has the potential to exceed 1.0, use the leading zero. If a value can never exceed 1.0, do not use the leading zero (per convention)
  • Table 2 - You can remove the No category
  • Table 3 - what does cut refer to

Discussion

  • Discuss the possible confounders and the possible explanations for the associations

Author Response

Abstract

  • Please provide n=, % in abstract

We have included the “n” in the result section (Line 25). This number is an unweighted number from the HINTS 5 Cycle 2 and 3. Since the percentage is the weighted percentage. We revised as “unweighted number = 1,327” in the result section of abstract (line 25).

Introduction

  • Justification of the choice of the three areas of this study (of the six) need to be explained and justified earlier in the paper

Our revised introduction provides the rationale for choosing the three areas of self-management behaviors (Line 65-91).

Material and Methods

  • You did not conduct a study but selected a cross-sectional slice of the data

We revised our methods (Line 100-105) as: “Health Information National Trend Surveys (HINTS) is a nationally representative cross-sectional survey administered by the National Cancer Institute with publicly available data.21 We conducted a secondary data analysis from HINTS 5 Cycle 2 and 3 that were performed before the coronavirus pandemic (i.e., 2018 to 2019). “

  • Flow diagramme is useful

Thank you.

  • Please clarified how some of the variables were operationalised. If you look at the questions - all of them could be done via the Internet email or phone - not necessarily mHealth apps? This present a major confounder in your study as the study associations imply causation but may be due to confounding.

We do recognize that self-management behaviors do not require the use of mHealth apps. However, in this study our objective was to assess the pattern in terms of cross-sectional association of mHealth apps use with self-management behaviors. For example, we believe that individuals who used mHealth apps may be more likely to use internet via smartphone, which in turn promote self-management behaviors.  We believe that mHealth apps use is one component of HIT use. We have revised our methods and discussion sections to clarify this point.

Methods (Line 192-197):

“It is possible that self-management behaviors can be achieved with the use of mHealth App or the use of other components of HIT such as email and internet.  In this study, we determine the association of mHealth app use with each of the self-management behaviors as mHealth App may facilitate use of other HIT components and further encourage self-management behavior.”

Discussion (Line 374-378):

“As the HINTS survey is a cross-section survey, we can only determine the association and unable to determine whether mHealth apps use caused better adoption of self-management behaviors (i.e., causality) “ 

  • Provide motivation for engagement classification

These three self-management behaviors are considered individual’s healthcare engagement as reported in the literature. We revised our method with the addition of references as the followings (Line 159-169):

“As all these three self-management behaviors are considered patients’ healthcare engagement, we further categorized individual different levels of healthcare engagement based upon their different self-management behavior involvement.25-28 Individuals who answered “no” to all three self-management behaviors were categorized as “no engagement” group. Individuals who answered “yes” to one self-management behavior was categorized as “low level of engagement”. Individuals with two self-management behaviors were categorized as “moderate level of engagement”. Ones with all three self-management behaviors were categorized as “high level of engagement”.

  • Justify age group categorization

This is a standard recode 5-level age categories used from HINTS. We revised and added the reference (Line 182).

  • In tables, please add the Test used and remove the leading 0s on the p-values.  If a value has the potential to exceed 1.0, use the leading zero. If a value can never exceed 1.0, do not use the leading zero (per convention)

We have done so.

  • Table 2 - You can remove the No category

We very much value the reviewer’s comment. However, we believe that including the number and weighted percentage of responders in “No” category will be more convenient for readers to understand and compare with “Yes” category. Therefore, it might interpret better on the association between the use of mHealth app and different self-management behaviors among individuals with/without the use of mHealth app. 

  • Table 3 - what does cut refer to

The different engagement level of self-management behaviors is a response variable in this ordinal logistic regression model, and it is a categorical data (ranging from 0 to 3). The cut-points shown in the Table 3 indicate where the different engagement level of self-management behaviors. These cut-pints are for the adjacent levels of the latent response variable which is a continuous data. Therefore, individuals who received a latent score less than 0.28 are classified as “no engagement level of self-management behavior”. Those who received a latent score between 0.28 and 1.72 are classified as “low engagement level”, ones who received a latent score between 1.72 and 2.92 are classified as “moderate engagement level” and ones great than 2.92 are classified as “high engagement level of self-management behavior”. In general, these are not used in the interpretation of the results. Please see detail explanation on website:  https://stats.idre.ucla.edu/stata/output/ordered-logistic-regression/

Discussion

  • Discuss the possible confounders and the possible explanations for the associations

We revised our discussion section to include the potential confounders that might not be addressed in this study and require further investigations in the future studies. Please see the following revisions (Line 374-384):

As the HINTS survey is a cross-section survey, establishing causation is not possible; in addition, multiple factors could affect individuals’ self-management behaviors. In this study, we only adjusted limited set of factors. It is possible that other factors (e.g., internet access, enrollment in life-style intervention programs, etc.) might potentially affect the use of mHealth app as well as self-management behaviors. Therefore, future studies are warranted to further investigate other potential risks affecting individuals’ self-management behaviors.”